# Lower Interstitial Glucose Concentrations but Higher Glucose Variability during Low-Energy Diet Compared to Regular Diet—An Observational Study in Females with Obesity

**DOI:** 10.3390/nu13113687

**Published:** 2021-10-20

**Authors:** Inger Nilsen, Agneta Andersson, Anna Laurenius, Johanna Osterberg, Magnus Sundbom, Arvo Haenni

**Affiliations:** 1Department of Dietetics and Speech Therapy, Mora Hospital, SE-792 51 Mora, Sweden; 2Center for Clinical Research Dalarna, Region Dalarna, SE-791 82 Falun, Sweden; 3Department of Food Studies, Nutrition and Dietetics, Uppsala University, SE-751 22 Uppsala, Sweden; agneta.andersson@ikv.uu.se; 4Department of Surgery, University of Gothenburg, SE-413 45 Gothenburg, Sweden; anna.laurenius@vgregion.se; 5Department of Surgery, Mora Hospital, SE-792 51 Mora, Sweden; johanna.osterberg@regiondalarna.se; 6Department of Clinical Sciences, Intervention and Technology (CLINTEC), Karolinska Institute, SE-171 77 Stockholm, Sweden; 7Department of Surgical Sciences, Uppsala University, SE-751 85 Uppsala, Sweden; magnus.sundbom@akademiska.se; 8Department of Public Health and Caring Sciences, Clinical Nutrition and Metabolism, Uppsala University, SE-751 85 Uppsala, Sweden; arvo.hanni@pubcare.uu.se; 9Department of Surgery, Bariatric Unit, Falun Hospital, SE-791 31 Falun, Sweden; 10Department of Diabetes, Endocrinology, University Hospital, SE-750 85 Uppsala, Sweden

**Keywords:** obesity, low-energy diet, dietary fiber, glycemic index, open-ended food record, continuous glucose monitoring, interstitial glucose, glycemic variability, mean amplitude of glycemic excursions, area under the curve

## Abstract

This is an observational study of interstitial glucose (IG) concentrations, IG variability and dietary intake under free-living conditions in 46 females with obesity but without diabetes. We used continuous glucose monitoring, open-ended food recording and step monitoring during regular dietary intake followed by a low-energy diet (LED). Thirty-nine participants completed both study periods. The mean BMI at baseline was 43.6 ± 6.2 kg/m^2^. Three weeks of LED resulted in a mean weight loss of 5.2% with a significant reduction in diurnal IG concentration but with greater glycemic variability observed during LED. The mean 24 h IG concentration decreased from 5.8 ± 0.5 mmol/L during the regular diet period to 5.4 ± 0.5 mmol/L (*p* < 0.001) during LED, while the mean amplitude of glycemic excursion increased from 1.5 ± 0.7 to 1.7 ± 0.7 mmol/L (*p* = 0.031). The positive incremental area under the curve at breakfast was significantly larger for LED compared to regular diet. The daily fiber intake and the glycemic index of breakfast meals were significantly associated with the glycemic variability during regular dietary intake. In conclusion, the 24 h mean IG concentration was lower but with more pronounced glycemic variability during LED compared to a regular diet.

## 1. Introduction

Obesity is a growing health problem worldwide and the prevalence of obesity has more than doubled since 1980 [1]. For patients with a body mass index (BMI) of 35 kg/m^2^ or above, bariatric surgery has become frequently used worldwide in the struggle against obesity-related complications such as type 2 diabetes and cardiovascular diseases [2,3]. In recent meta-analyses, bariatric surgery was shown to result in greater weight loss compared to non-surgical approaches [4], as well as a sustainable long-term weight loss [5].

Low-energy diet (LED) regimens, such as diet replacement formulas or hypocaloric diets, which are frequently used for weight loss, are also used immediately before bariatric surgery to reduce the risk of surgical complications [6]. Previous studies have demonstrated that LED resulted not only in lowered body weight and decreased liver volume, but also in lowered fasting and postprandial glucose concentrations and improved insulin sensitivity [7,8,9,10,11]. The Mediterranean diet demonstrated similar alterations, characterized by high contents of both monounsaturated fatty acids from olive oil as well as dietary fibers from fruits, vegetables and legumes [12,13]. Although the glycemic response of a mixed meal is influenced by the fat and protein content, the amount and type of carbohydrates have been suggested as the major contributors to the postprandial glycemic response [14]. Accordingly, diets higher in glycemic index (GI) or glycemic load were pointed out as causal factors contributing to the incidence of type 2 diabetes [15], while high carbohydrate quality diets (e.g., high in fiber, whole grains, low GI) were associated with lower prevalence of obesity and type 2 diabetes [16,17]. Furthermore, estimated dietary GI under free-living conditions was found to be a predictor of glycemic stability and variability in persons with type 2 diabetes [18] and an energy-reduced diet with low GI resulted in lowered glycemic variability in subjects with obesity but without diabetes [7]. Previous studies have demonstrated that LED regimens as well as the carbohydrate amount and quality may affect the glucose concentration, glycemic variability and health outcome. LED products such as total dietary replacement formulas are low in energy but might be proportionally high in carbohydrates, and even if weight reduction and improved fasting blood glucose and insulin concentrations are observed, the diurnal and postprandial glycemic variability has been sparsely explored in subjects without diabetes during this kind of regimen. As such, the aim of our study was to investigate diurnal and postprandial interstitial glucose (IG) concentrations in the everyday life of females with obesity but without diabetes, first during their regular dietary intake and then during LED treatment. Our secondary aim was to study whether there were significant relationships between dietary factors and physical activity level on the one hand and glucose concentrations or glycemic variability on the other.

## 2. Materials and Methods

### 2.1. Study Design and Participants

This is a prospective observational study of female subjects with BMI ≥ 35 kg/m^2^ in a free-living setting before bariatric surgery. Since almost 80% of patients treated with bariatric surgery in Sweden are females [19], we chose to include only female subjects in this study. We measured interstitial glucose concentrations during the participants’ regular food intake (study period 1) and after a median of three weeks of LED treatment (study period 2). Continuous glucose monitoring (CGM), open-ended food recording and a step monitor measuring step counts were used for four days in each study period (Figure 1). Data for the last three days in each study period were analyzed due to methodological issues with the CGM [20]. Before the study started, all patients underwent a medical assessment, including laboratory tests at accredited laboratories (fasting concentrations of blood glucose, HbA1c and insulin).

Out of 104 eligible females, 45 were recruited for study period 1, and of these 39 (86.6%) returned for study period 2 (Figure 1). Exclusion criteria were diagnosis of diabetes mellitus, the use of medications affecting glucose metabolism or bowel peristalsis, previous bariatric surgery, celiac disease or milk protein intolerance.

### 2.2. Continuous Glucose Monitoring

To be able to study glucose concentrations and glycemic variability continuously during the everyday life of the study participants, we used the Minimed IPRO-2 CGM system (Medtronic, Northridge, CA, USA) for measuring of glucose concentrations in the extracellular interstitial fluid (i.e., every ten seconds) to present the mean glucose value every five minutes. IG concentrations are highly correlated to the circulating blood glucose concentrations, with a latency time of five to fifteen minutes [20].

At the start of each of the two study periods, participants attended outpatient units at a community hospital and a university hospital in Sweden, where the CGM thin-wire sensor was inserted in accordance with the manufacturer’s instructions. Weight and height were measured at the same visit. For calibration purposes, patients were instructed to measure their own capillary blood glucose with the Bayer Contour Next One glucose meter (Ascensia Diabetes Care Holdings AG, Basel, Switzerland) four times daily (before breakfast, lunch, dinner and at bedtime). Data were uploaded to the Carelink software program (Medtronic, Northridge, CA, USA) and then exported to Microsoft Excel 2016 (Microsoft Corporation, Redmond, WA, USA).

We analyzed the 24 h mean glucose concentration and mean amplitude of glycemic excursions (MAGE), as well as the postprandial mean glucose concentration and the positive incremental area under the curve (iAUC) after breakfast. MAGE was calculated according to Hill et al. using EasyGV version 9.0 (Nathan RJ Hill, University of Oxford, Oxford, UK) with a cut-off level for high MAGE of ≥2.8 mmol/L [21]. The trapezoid rule was used to calculate the 120 min positive iAUC after breakfast [22]. We chose to analyze the postprandial glucose response in connection with breakfast to minimize potential second meal effects [23]. Following the guidelines of the International Diabetes Federation and the American Diabetes Association, we used 7.8 mmol/L and 3.9 mmol/L as cut-off levels for high and low glucose concentrations [24,25].

### 2.3. Dietary Intake and Dietary Methods

During the first study period, the subjects were instructed to continue with their regular diet. For the second period, all participants received a LED regimen containing 800–1100 kcal/d. The recommended LED consisted of a powder-based total diet replacement product for weight control (Modifast^®^, Impolin AB, Taby, Sweden), which after preparation with water was consumed as a liquid or semisolid meal (shake, soup, porridge, pudding and pasta). The LED was distributed free of charge. The energy and macronutrient content of the LED product was 220 kcal, fat 4.5 g, carbohydrate 27 g, fiber 4.2 g and protein 14.3 g per portion. The patients were instructed to take 4–5 portions distributed over the day. Except for an additional 1.5–2 L of non-caloric fluid, the patients were asked not to eat anything else during the 4-week LED period.

During both study periods, the participants documented their entire intake of food, LED products and fluids in a paper-based, open-ended food record, giving the starting times of the meals. Dietary intake was self-reported in household measures and we obtained detailed descriptions of recipes, food brands and food content [26]. We calculated the absolute daily amounts of energy and macronutrients as well as those for all the breakfast meals using the Dietist Net Pro 2020 (Diet and Nutrition Data AB, Bromma, Sweden) software program and the Swedish Food Composition Database (version 20200116) plus food composition databases from the food industry. Breakfast was defined as the first meal of the day containing at least 50 kcal [27].

The GI for each food item was assessed according to the international GI tables presented by Atkinson et al. [28] and was based on the glucose scale. We calculated the contribution of each food item to the total amount of carbohydrates for each of the breakfast meals, and thereafter the GI for the entire breakfast meals, according to WHO/FAO guidelines [29]. Meal GI was calculated by the first author (I.N.) and thereafter checked and discussed with the two other dietitians (A.L. and A.A.). Similar to the latest Swedish National Survey of Dietary Intake in Adults, we calculated the fiber intake in eight major food categories and compared the total dietary fiber intake to the nutrition recommendations [30,31].

### 2.4. Physical Activity

An ActiGraph^TM^ wGT3X-BT activity monitor (Pensacola, FL, USA) was used to record step counts during the two study periods [32]. The patients wore the device around the waist according to the manufacturer´s instructions. After each study period, data were uploaded to the Actilife program version 6.13.3 and only data with a minimum of 9 h daily wear time were included in the analysis.

### 2.5. Statistical Analysis

Because three days of CGM data were analyzed for each study participant, we performed a linear mixed model analysis to examine the possible influence of the first, second and third days on the IG indices. Since the day of measurement had no influence on the glucose variables in either of the test periods, all days were used in the main data analysis.

Every 24 h period produced 288 glucose values for each person, and we calculated the 24 h mean values per person and day for the IG indices. Subsequently, the mean IG was calculated for the entire group. The same principles were used for the breakfast analysis. To compare data between the two study periods, we used paired data samples i.e., data from the 39 females participating and completing both study periods.

Linear regression analysis was performed to examine any relations between age, BMI, step counts, breakfast GI and the energy-adjusted macronutrient intake on the one hand and the interstitial glycemic response indices on the other. Independent variables that were found to relate significantly to the glycemic indices in the simple regression analysis were added stepwise to the multiple regression model. Since independent observations represent one of the assumptions for linear regression models, mean values for the three days for each of the 45 participants in study period 1 were used in the analysis. Examination of the model residuals supported that the assumptions of linearity, homoscedasticity and normality were met [33]. Diurnal and breakfast variables for the regular dietary intake were analyzed separately in this manner, but not the LED intake, since the participants had similar nutritional intake during this period.

The results are presented as mean values and standard deviations (SD), and we used paired sample t-tests for comparison of data with normal distributions. For data with non-normal distributions, we used the median and interquartile range (IQR) and the Wilcoxon signed rank test to compare median values. The McNemar test for related samples was used to compare glucose concentrations below or above the defined cut-off levels for the regular diet period and the LED period. The level of significance was 2-sided (*p* < 0.05) for all analyses. We used IBM SPSS Statistics for Windows version 26.0 (IBM Corp: Armonk, NY, USA) for the statistical analysis.

## 3. Results

### 3.1. Characteristics of Study Participants

The baseline characteristics for the 39 females participating in both study periods are shown in Table 1. After a median period of three weeks of the four week LED regimen, the mean weight loss was 6.4 ± 2.6 kg, which corresponded to 5.2% total body weight loss and a BMI of 41.4 ± 5.7 kg/m^2^ (*p* < 0.001).

### 3.2. Diurnal IG Concentrations

All but three of the 39 participants had three days each of CGM data, which resulted in a total of 114 days being included in the analysis. Our results show that the mean IG concentration was significantly lower during the LED treatment compared to the regular diet period (Table 2, Figure 2). LED resulted in even lower glucose concentrations during the night (00:00–04:00) compared to regular diet, with 5.0 ± 0.8 and 5.8 ± 0.9 mmol/L, respectively (*p* < 0.001) (Figure 2). Conversely, the glycemic variability represented as MAGE was significantly higher during the LED period, while low glucose concentrations (≤3.9 mmol/L) were more frequent during this period (Table 2).

### 3.3. Breakfast IG Concentrations

One person omitted breakfast for one day during the LED period; hence, 113 days were included in the breakfast analyses. The mean of 120 min postprandial IG concentrations after breakfast was similar for both study periods (Table 3, Figure 3A); however, the postprandial glucose increment was larger for the LED regimen compared to the regular diet, as was the positive iAUC (Table 3, Figure 3B).

### 3.4. Dietary Intake and Physical Activity

The absolute amount of reported daily and breakfast intakes of energy and macronutrients was significantly lower during the LED regimen compared to the regular diet period, except for fiber intake, which was similar during both periods (Table 4). According to the reported food intake during the LED regimen, 16 of 114 days contained a limited amount of food items in addition to the LED products. All but one of these 16 days had total energy intakes below the upper limit of the prescribed LED regimen of 1100 kcal/d. The median number of daily LED portions was 4.

During the LED regimen, the daily energy percentages (E%) for carbohydrate, fat and protein were 51%, 21% and 28%, respectively. The corresponding distributions for the regular diet period were 40%, 42% and 16%, respectively, in addition to 2 E% from alcohol. The E% for breakfast intake was similar to the diurnal distribution for both study periods but with no alcohol.

The main sources of daily fiber intake during the regular diet period were bread; rice; pasta and cereals (32%); and vegetables, fruit and potatoes (31%) (Figure 4); however, on only 12 of the 114 days did fiber intake reach the level of nutritional recommendations of at least 25 g per day [30]. The LED products contributed 95% of the fiber intake during the LED period, in which inulin was by far the most dominant fiber source. None of the study participants reported taking any fiber supplements in either of the two study periods.

Physical activity, assessed by step counts, was similar during the two periods (6654 ± 2571 vs. 6519 ± 2998 steps per day, *p* = 0.699). Out of 114 days during each study period, 23 days from period 1 and 22 days from period 2 were excluded due to a daily registration time of less than 9 h. This implies 91 and 92 days with at least 9 h registration of step counts and with 78 pairs of days for statistical comparison.

### 3.5. Independent Contributors to the Glycemic Response

The linear regression analysis for the 24 h period revealed significant associations between daily energy-adjusted fiber and protein intake and MAGE during regular diet, with only fiber reaching the level of significance in the multiple regression analysis (Table 5). The R-square values indicated that fiber and protein intake explained 22% of the variation in MAGE. None of the remaining variables were significantly related to the 24 h IG response.

In the linear regression analysis for the breakfast meal during regular diet, we found a positive association between meal GI and positive iAUC (unstandardized β = 0.87, *p* = 0.032, 95% CI (0.08, 1.65)). Figure 5 shows the scatterplots of the relationships between daily fiber intake and MAGE (A) and the relationship between breakfast GI and the positive iAUC (B).

In addition, age was positively related to the mean breakfast glucose concentration (β = 0.02, *p* = 0.047, 95% CI (0.00, 0.04)), which implies that higher age was associated with higher postprandial glucose concentrations. None of the remaining variables were significantly related to the IG response at breakfast.

## 4. Discussion

Our main findings were that three weeks of LED resulted in a median weight loss of 5.2% with a significant reduction in diurnal mean interstitial glucose concentration and with a more pronounced reduction of the mean nocturnal glucose concentrations, but with a higher glycemic variability observed during the LED. Moreover, we observed a negative association between the energy-adjusted daily fiber intake and MAGE, as well as a positive relationship between the breakfast GI and positive iAUC during the period of regular dietary intake.

The reduction of diurnal mean IG concentrations during LED found in this study is in line with results from previous studies examining alterations in fasting blood glucose concentrations after LED treatment [10,34]. The ameliorated glucose concentrations might be attributed to an improved insulin sensitivity as well as the lower carbohydrate intake during LED [9,34]. In spite of the lower mean diurnal IG concentration, we observed significantly higher 24 h MAGE during the LED period compared to the period with regular dietary intake; however, during both study periods, mean MAGE values were below the suggested upper normal range of 2.8 mmol/L [21]. Since high glycemic variability has been associated with increased oxidative stress and impaired vascular endothelial function [35,36], it is worth noting that in 7% of the days during the LED regimen and in 4.4% of the days during regular diet, MAGE was above this upper reference level. Furthermore, we observed instances of diurnal glucose concentrations ≥7.8 mmol/L for about one-third of the days during both study periods and instances of glucose concentrations ≤3.9 in 22% and 9% of the days during LED and regular diet, respectively.

The observations in our study might be compared to those reported by Buscemi et al., showing lower glycemic variability and improved endothelial function with an energy-reduced diet with low GI [7]; hence, the unexpected larger positive iAUC for the postprandial IG response after breakfast during the LED period might be explained by the rapidly digestible carbohydrates in the LED product used in our study. The most common carbohydrate sources were maltodextrin, skimmed-milk powder and glucose syrup, besides inulin [37]. The iAUC after breakfast during the LED period in our study was comparable to the results reported by Vrolix et al., whereby healthy adults ingested a fruit drink containing 25 g of carbohydrates from sucrose but without any fat or protein [38]. Moreover, the LED used in our study was lower in energy compared to the regular diet and consisted of liquid or semisolid meals, which have previously been found to result in a more rapid gastric emptying rate [39,40]. This might have resulted in accelerated absorption of the carbohydrates in the LED breakfast.

The mean fiber intake during regular diet was low, with only 10% of the days reaching the recommended level of at least 25 g of fiber per day [30]. Furthermore, we showed that fiber intake was negatively associated with MAGE, implying that increased fiber intake might lower the diurnal glycemic variability. This is consistent with previous studies reporting that an increased fiber intake reduced the postprandial blood glucose concentration and the diurnal glycemic variability in subjects with overweight or obesity but without diabetes [41,42]. The effects of dietary fiber intake on the glycemic response is probably elicited through several mechanisms, such as slowed digestion and absorption of glucose by certain soluble fiber types, as well as the bacterial degradation of indigestible fibers, which might affect insulin sensitivity [43,44]. We found that at least one-third of the total dietary fiber intake during regular dietary intake came from food rich in soluble fiber such as fruits, vegetables and legumes, while a similar proportion came from common sources of insoluble fiber such as bread, rice, pasta and cereals. Moreover, we observed a wide range of GI levels in breakfast meals and a significant relationship between GI and the postprandial positive iAUC during regular dietary intake. This is compatible with the findings by Kochan et al., whereby GI was found to be an important determinant of glycemic response after self-selected breakfast meals in subjects with obesity [45]. In addition, Lagerpush et al. showed that a diet with a GI level of 40 compared to a diet with a GI level of 74 resulted in significantly lower diurnal iAUC [46]. Indeed, previous studies have pointed out that a dietary pattern rich in high-fiber or low-GI food items such as fruits, vegetables and whole grains is helpful in reducing the risk of obesity and type 2 diabetes [16,17]; hence, the low fiber intake found in our study group during their regular dietary intake is of concern from a public health perspective.

One strength of our study is its prospective design, with the study participants being their own controls. Another strength is the homogeneity of the participants in the study regarding gender, absence of diabetes, not using medications with known glucometabolic or gastrointestinal effects and with all patients carefully investigated and approved for bariatric surgery. Moreover, the use of the CGM method, which continuously measures glucose concentrations under free-living conditions, enabled detection of alterations in glucose status that are difficult to discover by self-testing of blood glucose or in a laboratory setting [20]. A further strength is the consideration of the degree of physical activity measured by a step monitor. Regarding our study’s potential limitations, the comparison of LED with regular diet was not ideal because of the different textures with these two diets; however, we wanted to investigate the most common type of LED treatment before bariatric surgery in a real life observational setting, although being aware that various LEDs with different textures might give different results. Furthermore, the self-reported data for dietary intake should be interpreted with caution due to the risk of underreporting, which is especially common among people with a higher BMI [47,48]. The reported energy intake during the period of regular dietary intake was low when taking the estimated basal metabolic rate with an energy-balanced state into account; however, during the LED treatment, the reported mean energy intake was within the recommended diet regimen, although at the lower end of the interval prescribed. To prevent incorrect reporting, all participants were fully guided individually by a dietitian with extensive experience in obesity care in how to report their food intake in detail regarding contents and time. Upon return, all food records were carefully checked along with the study subjects so that ambiguous recording could be clarified. We think that these efforts to properly register food intake constitute one of the strengths of this study.

## 5. Conclusions

In conclusion, we found that three weeks of treatment with a low-energy diet compared to a regular diet resulted in a significantly lowered diurnal mean interstitial glucose concentration but greater glucose variability in females with obesity but without diabetes mellitus. Furthermore, our observations from the regular diet period suggest that an increased daily intake of dietary fiber and a lower glycemic index of breakfast meals might lower the glucose variability.

With reference to these findings, we suggest that future bariatric studies take the food texture and carbohydrate quality into account when investigating the glucometabolic effects of hypocaloric diets, as this might be of particular importance in long-term treatment with hypocaloric regimens.

## Figures and Tables

**Figure 1 nutrients-13-03687-f001:**
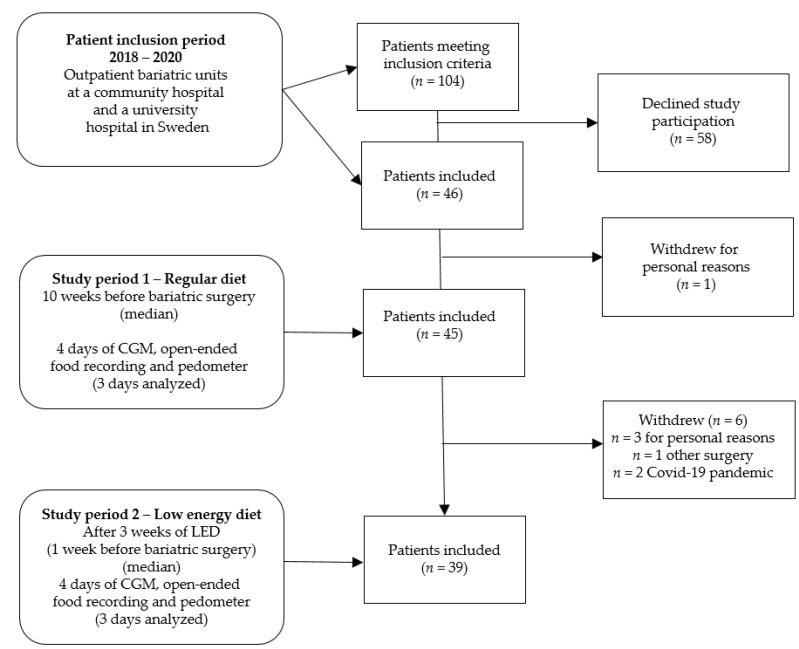
Flow chart of the study participants and the study process. CGM = continuous glucose monitoring; LED = Low-energy diet.

**Figure 2 nutrients-13-03687-f002:**
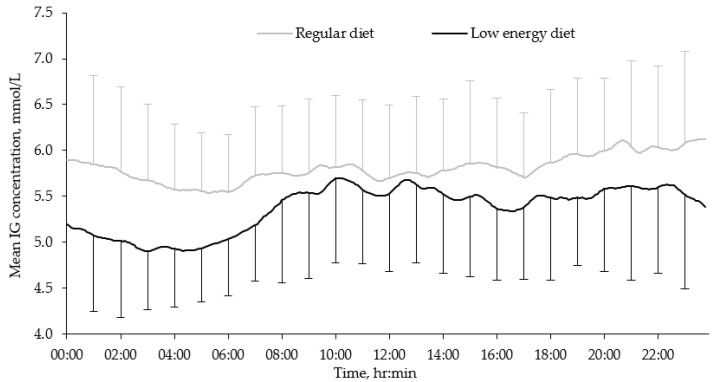
Mean diurnal interstitial glucose concentrations during regular diet and LED. Mean values of glucose from 114 days in 39 females with obesity were plotted every 5 min; means + SD for regular diet and – SD for LED were plotted every 60 min. LED = low-energy diet; IG = interstitial glucose.

**Figure 3 nutrients-13-03687-f003:**
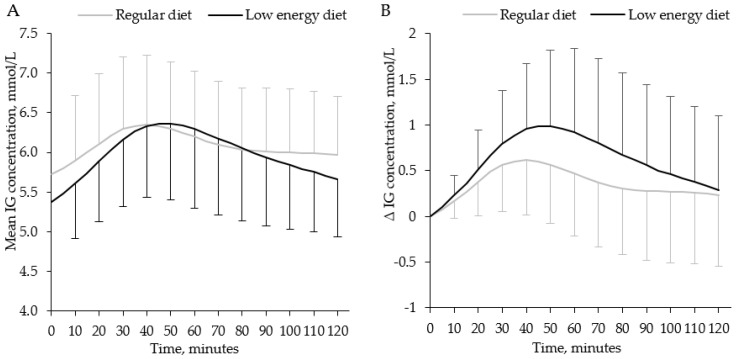
(**A**) Mean postprandial interstitial glucose concentrations and (**B**) mean increments of interstitial glucose concentrations 0–120 min after breakfast during regular diet and LED. Mean values of glucose from 113 breakfast meals in 39 females with obesity were plotted every 5 min and SD were plotted every 10 min. LED = low-energy diet; IG = interstitial glucose.

**Figure 4 nutrients-13-03687-f004:**
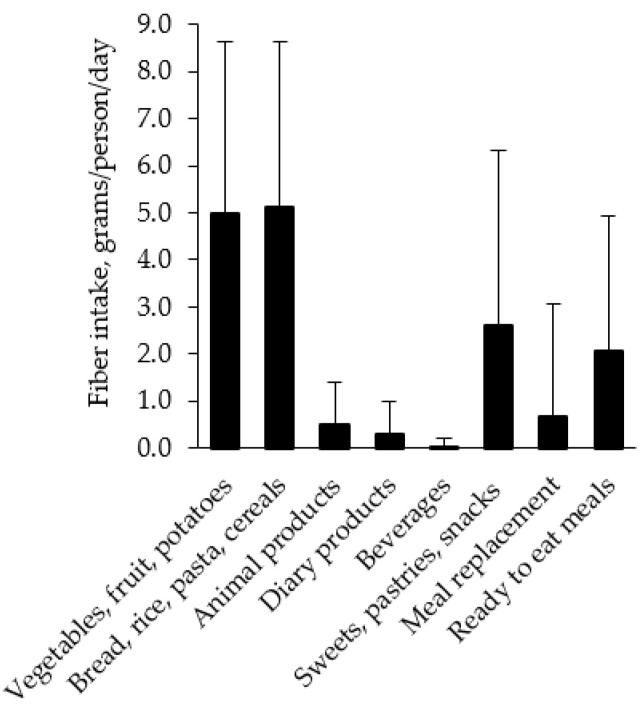
Distribution of daily fiber intake between different food groups during regular dietary intake. Means + SD for 114 days in 39 women.

**Figure 5 nutrients-13-03687-f005:**
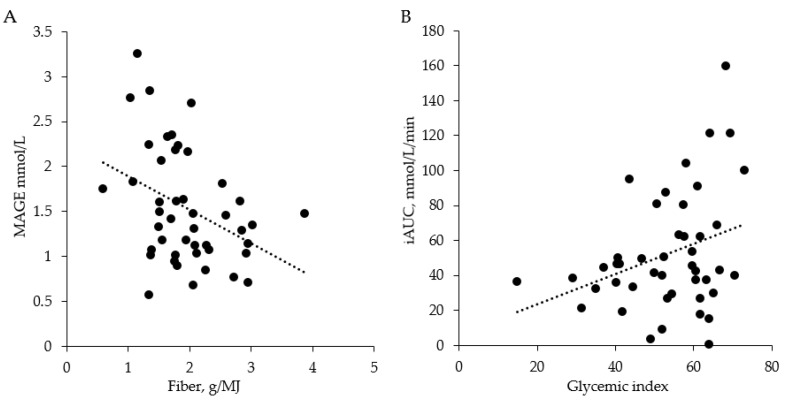
Scatterplots showing the relationship between (**A**) diurnal fiber intake and MAGE and (**B**) that between breakfast GI and the positive iAUC during regular dietary intake. Dots are based on means of three-day measurements for the 45 study participants included in study period 1. GI = glycemic index; iAUC = incremental area under the curve; MAGE = mean amplitude of glycemic excursions.

**Table 1 nutrients-13-03687-t001:** Baseline characteristics of the 39 female participants ^1^.

	Patient Characteristics
Age ^2^, y	37 (17)
Body weight, kg	122.5 ± 20.3
BMI, kg/m^2^	43.6 ± 6.2
Fasting P-glucose, mmol/L	5.7 ± 0.5
B-HbA1c, mmol/mol	36.6 ± 4.0
Fasting S-insulin, mE/L	25.8 ± 1.5

^1^ Results are presented as means ± SD unless otherwise indicated. ^2^ Median (IQR). HbA1c = glycosylated hemoglobin; BMI = body mass index.

**Table 2 nutrients-13-03687-t002:** Diurnal interstitial glucose variables for 114 days in 39 females during the regular diet period and the LED period ^1^.

	Regular Diet	LED	*p*
24 h mean glucose, mmol/L	5.8 ± 0.5	5.4 ± 0.5	<0.001
24 h MAGE, mmol/L	1.5 ± 0.7	1.7 ± 0.7	0.031
Glucose ^2^ ≥ 7.8 mmol/L	38 (33%)	31 (27%)	0.229
Glucose ^2^ ≤ 3.9 mmol/L	10 (9%)	25 (22%)	0.011
MAGE ^2^ ≥ 2.8 mmol/L	5 (4%)	8 (7%)	0.581

^1^ Results are presented as means ± SD from paired sample t-test (2-tailed) unless otherwise indicated. ^2^ Number of days (proportions) with instances of glucose above or below defined cut-off levels and McNemar’s test for related samples (2-tailed). LED = low-energy diet; MAGE = mean amplitude of glycemic excursions.

**Table 3 nutrients-13-03687-t003:** Two-hour postprandial interstitial glucose variables for 113 breakfast meals in 39 females during the regular diet period and the LED period ^1^.

	Regular Diet	LED	*p*
Mean of 0–120 min glucose, mmol/L	6.1 ± 0.7	6.0 ± 0.7	0.209
Mean glucose at 0 min, mmol/L	5.7 ± 0.8	5.4 ± 0.6	<0.001
Mean glucose at 120 min, mmol/L	6.0 ± 0.7	5.7 ± 0.7	<0.001
120 min positive iAUC ^2^, mmol/L/min	43.5 (43.6)	77.8 (62.4)	<0.001
Glucose ^3^ ≥ 7.8 mmol/L	13 (12%)	16 (14%)	0.648
Glucose ^3^ ≤ 3.9 mmol/L	0	0	-

^1^ Results are presented as means ± SD from paired sample t-test (2-tailed) for the entire 120 min postprandial period and for the glucose concentrations at 0 min and 120 min after breakfast. ^2^ Median (IQR) and Wilcoxon signed rank tests (2-tailed) for the positive iAUC for the entire 120 min postprandial period. ^3^ Number of days (proportions) of breakfast meals with instances of glucose above or below defined cut-off levels and McNemar’s test for related samples (2-tailed). iAUC = incremental area under the curve; LED = low-energy diet.

**Table 4 nutrients-13-03687-t004:** Mean daily energy and macronutrient intakes for 114 days in 39 females (113 days for breakfast data) during regular diet and LED ^1^.

	Regular Diet	LED	*p*
Daily intake:			
Energy, kcal	2069 ± 688	827 ± 197	<0.001
Carbohydrate, g	195.3 ± 7.0	98.5 ± 24.7	<0.001
Fiber, g	16.1 ± 6.1	16.5 ± 4.9	0.583
Fat, g	97.9 ± 42.0	19.4 ± 9.2	<0.001
Protein, g	79.7 ± 24.3	56.6 ± 11.1	<0.001
Alcohol, g	7.6 ± 18.7	0.0	<0.001
Days on LED ^2^	-	21 (2)	-
Breakfast intake:			
Breakfast time ^2^, h:min	8:30 (2:45)	9:00 (2:00)	0.006
Energy, kcal	401 ± 198	218 ± 26	<0.001
Carbohydrates, g	42.2 ± 21.7	26.4 ± 5.2	<0.001
Fiber, g	3.8 ± 2.6	4.2 ± 1.2	0.088
Fat, g	17.6 ± 11.5	4.9 ± 1.3	<0.001
Proteins, g	16.9 ± 8.9	14.8 ± 3.0	0.026
GI ^3^	53 ± 15	-	<0.001

^1^ Dietary data based on a 3-day self-reported food record and presented as means ± SD from paired sample t-test (2-tailed) unless otherwise indicated. ^2^ Median (IQR) and Wilcoxon signed rank tests (2-tailed). ^3^ International tables of glycemic index (28). GI = glycemic index; LED = low-energy diet.

**Table 5 nutrients-13-03687-t005:** Results of linear regression analyses performed to determine independent contributions to the diurnal interstitial glycemic response during the regular diet period ^1^.

	Simple Linear Regression of 24 h Glycemic Response				Multiple Linear Regression of 24 h Glycemic Response		
	Mean Glucose, mmol/L		MAGE, mmol/L		MAGE, mmol/L		
	β	P	β	P	β	P	95% CI
Age, y	0.01	0.103	0.00	0.938			
BMI, kg/m^2^	−0.00	0.724	0.02	0.303			
Carbohydrate, g/MJ	0.01	0.500	0.01	0.634			
Fiber, g/MJ	−0.21	0.064	−0.37	0.011	−0.31	0.032	−0.59, −0.03
Protein, g/MJ	−0.05	0.321	−0.15	0.017	−0.12	0.052	−0.24, 0.00
Fat, g/MJ	−0.02	0.722	0.04	0.546			
Step Counts	0.00	0.159	0.00	0.766			

^1^ Results are based on means of three days for the 45 study participants included in study period 1 and presented as unstandardized β and 95% CI. MAGE = mean amplitude of glycemic excursions.

## Data Availability

Data and data collection tools described in the manuscript will be made available upon resonable request.

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
