# Peer review of "Lower Interstitial Glucose Concentrations but Higher Glucose Variability during Low-Energy Diet Compared to Regular Diet—An Observational Study in Females with Obesity"

_nutrients, 2021, doi:10.3390/nu13113687_

Round 1

Reviewer 1 Report

The paper describes a small portion of a larger study, where the glucose response to a typical and low-energy diet was recorded in obese females intended to undergo bariatric surgery. The results were interesting in that an increase in glucose fluctuation was unexpectedly found in response to the prescribed weight-loss diet. However, the paper fails to explain why this small analysis needs to be addressed separately, rather than leaving it as part of the main study.

The introduction provided a good background overview, though perhaps more detail should be given to discuss bariatric surgery if it is an essential part of the premise. How common is the surgery? Is it the best weight-control measure, and if so why? Additionally, more explanation is needed on the study rationale; for instance, why were only female subjects chosen?

[Line 35] Is there a reason that 'Conclusion' is the only sub-heading included in the abstract?

[Line 83] What is this larger study, and how does this study fit in?

[Line 86] Was the data from the 6 participants who did not return for study 2 used for study 1?

[Line 96] Is there a reason interstitial glucose was measured instead of circulating levels?

[Line 119-120 and 124] Was the exact intake, that is number of portions, per day recorded?

[Line 134] Was the time of breakfast and other meals recorded or analysed?

[Line 149-150] Were there many days that had to be excluded for less than 9 hours usage?

[Line 167-168] What was meant by this sentence? What were the other assumptions and how were they fulfilled?

[Line 321] Does GV stand for glucose variability? This abbreviation has not been defined or used anywhere else in the manuscript.

Reviewer 2 Report

The main aim of this study was to investigate diurnal and postprandial interstitial glucose concentrations in the everyday life of females with obesity, but without diabetes, first during their regular dietary intake and then during LED treatment.

The aim of the study was achieved and the results obtained can effectively help in the fight against obesity.

The research were properly planned and conducted. From the merits point of view, the work does not raise any objections.

Please find below my minor comments and suggestions:

  • Why weren't study participants asked if they were taking dietary supplements?
  • Figure 4 shows Distribution of daily fiber intake between different food groups during regular dietary intake. In my opinion, it would be beneficial to include data on this chart for the LED diet as well.

Reviewer 3 Report

This study is interesting with comparing the effect of regular diet and low energy formula diet (ie, energy consumption). The primary outcome was the interstitial glucose concentration during the 24h period and after the breakfast. The author concluded that LED result to higher GV compared to the regular diet. Secondary outcome was evaluating the relationship between macronutrient amount and the diurnal interstitial glycemic response. This study contains major problem in the study settings. The fluidity of the both diet was completely different which attribute to all the postprandial glycemic response. Based on the different fluidity of two diets (solid versus liquid/semiliquid), we could not provide the result from macronutrient volume nor the amount of the fiber.

Major

  1. In figure 3, authors reported the delta-IG concentration increased after the breakfast. However, this difference might be the result from the meal fluidity. Liquid formula shows rapid glucose excursion than the solid meal. LED should be the solid low energy food, instead of the liquid formula.

Round 2

Reviewer 3 Report

Thanks for the authors. They have fully recognize the problem and have been improved with additional revised version.